# Paediatric Care in The Netherlands: State of Affairs, Challenges and Prospects

**DOI:** 10.3390/ijerph19031037

**Published:** 2022-01-18

**Authors:** Danielle Jansen, Károly Illy

**Affiliations:** 1Department of General Practice, The University Medical Center, University of Groningen, 9713 GZ Groningen, The Netherlands; 2Department of Sociology, University of Groningen, 9712 TG Groningen, The Netherlands; 3Department of Pediatrics, Hospital Rivierenland, 4002 WP Tiel, The Netherlands; voorzitter@nvk.nl; 4Dutch Paediatric Society, 3528 BL Utrecht, The Netherlands

**Keywords:** paediatric care, quality improvement, organisation of care, The Netherlands

## Abstract

There are many societal developments in The Netherlands, such as a rising and changing demand for care and support and a paradigm shift from curation to prevention, that currently—and in the near future—will have an impact on paediatric care. These developments both reveal potential risks in paediatric care and affect practices that require future improvement. In this viewpoint, we first present the most pressing developments for paediatrics, and we demonstrate why and how Dutch paediatricians have renewed their vision on paediatric care in order to cope with a changing society. It is a vision towards the year 2030 that gives children and paediatric care the right place in the Dutch healthcare landscape to guarantee accessible, high-quality, and effective care for every child at the right time. Realising this renewed vision requires however not only an adjustment from paediatricians and paediatricians in training, but also from professionals who work with them and from the Government that can facilitate or accelerate the implementation of the renewed vision in various ways.

## 1. Introduction

Paediatric care in The Netherlands deals with the treatment of children aged 0 to 18 years. In order to receive paediatric treatment, children need to consult a GP first, who will provide a referral letter. On 1 January 2020, The Netherlands counted 3.3 million children, more than 19% of the total population of The Netherlands [1]. Of these children, an estimated number of 385,848 children (11.7%) visited a paediatrician in 2020. According of the website of the Dutch Healthcare Authority, which provides care and treatment data supplied by Dutch hospitals (e.g., how many patients are diagnosed, the associated care products and care activities), this number is lower than previous years (452,138 in 2019 and 447,081 in 2018) because of the COVID-19 pandemic [2]. As in other countries, regular paediatric care in The Netherlands has been scaled down and fewer children turned to the emergency room, for e.g., traffic accidents and viral infections. See Table 1 for child health indicators in The Netherlands.

In recent years, the reasons for consulting a paediatrician have remained roughly the same. The fixed top three of consultations in 2018, 2019 and 2020 concerned basic care for newborns (18.5%, 18.6% and 20.5%, respectively), followed by asthma care (7.1%, 6.8% and 6.4%, respectively) and follow up neonatal issues (non-NICU) (3.9%, 3.7% and 4.1%, respectively). The following conditions do appear in the further top 10 across all three years, albeit in different positions: upper respiratory infections, constipation, chronic recurrent abdominal pain, food allergy, short stature/deflecting height growth curve, co-treatment: nutrition/hydration policy/antibiotics, pain relief (2018 and 2020). In addition to this stability in complaints and disorders with which children present themselves to the paediatrician, changes can also be noted: whereas lower respiratory infection were presented in 2–2.1% of the children in respectively in 2018 and 2019. Further, it only concerned 1.3% of the children in 2020. The same goes for gastroenteritis, which was responsible for 2.4–2.9% (2018 and 2019, respectively) and for only 1.3% in 2020. Adiposity was a condition that did not appear in the top 10 of 2018 and 2019 (both 1.6%) but did appear in the top 10 of 2020 (1.8%) [2].

There are many societal developments in The Netherlands that currently and in the near future have an impact on Dutch health care in general and on paediatric care in particular. These developments both reveal potential risks in paediatric care and affect practices that require future improvement. The Dutch Paediatric Society anticipates this by aligning its vision on paediatric care with current and future developments. In this viewpoint, we first present the most pressing developments for paediatrics, and we demonstrate why and how paediatricians need to break new ground in order to cope with a changing society.

## 2. Trends and Developments That Require a Different Focus and Organisation of Paediatric Care

### 2.1. A Rising and Changing Demand for Care and Support

The demand for care in The Netherlands is changing because of several social developments of which we will highlight three important ones: an aging population, more diversity in cultural background and a rising prevalence of chronic diseases and multimorbidity among children and adolescents. 

Similar to other industrialised countries, also The Netherlands is experiencing prolonged life expectancies and an ageing population. The life expectancy at birth for males and females in The Netherlands is one of the highest in the world and continues to increase every year. In 2019, the life expectancy for males in The Netherlands was 80.5 years and 83.6 years for females [10]. For comparison, in 1994 it was 74.6 years for men and 80.3 years for women [11]. This changing demography causes a changed age distribution of patients who present to the general practitioner (GP). Some GPs are seeing a smaller number of children in their overall patient population that may be necessary to achieve competence and confidence in paediatric care. Consequently, this smaller proportion of children in GP practice may have led to an increase in general practitioner-referred patients to paediatricians. This may increase the likelihood that the paediatrician will now provide more care to a pediatric population previously treated in primary care.

In addition to an aging population, a more culturally diverse population also causes a rising and changing demand for paediatric care and support. In 2020, almost 28% of all 0 to 25-year-olds in The Netherlands had a migration background. This group consists of more than 66% children and youngsters with a non-Western migration background (i.e., Moroccan, Turkish, Surinamese, Syrian, Afghan, Iranian and Iraqi background) [12]. Children and adolescents with a non-Western migration background bring different disease prevalence, habits, values, and norms with them. For example, overweight and obesity are significantly more prevalent in young migrant children of non-Western descent, as compared to native children and children of Western descent [13]. However, the migrant background in this young population is also accompanied by language differences and insufficient patient, parents and guardians health literacy, factors that can challenge patient–physician communication and require new capacities from the paediatrician [14].

The third cause of a rising and changing demand for paediatric care and support which we wish to emphasise is the rising prevalence of chronic diseases and multimorbidity among children and adolescents. A Dutch study showed that in 2018, over 1.3 million children and young people aged 0–25 years in The Netherlands (more than 1 in 4) deal with a chronic condition [15]. It concerns various diseases such as diabetes, rheumatism, cystic fibrosis, asthma and eczema but also mental disorders such as ADHD and depression. There is an increase for most chronic conditions when compared to previous years. For example, in 2020, nearly 4% of young people aged 12 to 18 say they experienced self-reported feelings of depression for at least six months in the past year; in 2014 this was 1.8% [16]. An increasing prevalence of chronic diseases in children not only asks for a preventive role for the paediatrician, rather than a curative role, it also requires more collaboration with children and parents themselves and with public health professionals. In order to provide the right care and with that, to establish a reduction in chronic diseases, it is desirable that paediatricians are an integral part of preventive efforts [17]. 

### 2.2. Towards an Ideal and Flexible Doctor–Patient Collaboration

Besides a rising and changing demand for care and support because of several demographic and social developments, a changing doctor–patient relationship is also a continuous development that require a different focus and organisation of paediatric care. Easy access to medical information and patient records has changed and challenged the relationship between paediatrician and patient and his/her family. This change can be observed throughout the medical world: it is not the doctor anymore who takes a unilateral decision (paternalism), but the decision is taken in collaboration between doctor and patient and with the explicit consent of the patient. However, not every patient is suitable for or charmed by shared decision making; if one as a patient or parent of a patient lacks the necessary knowledge and skills to make treatment decisions, then one is more inclined to leave the decision-making to the doctor. Therefore, today’s paediatrician is required to have a flexible attitude in which he/she constantly customises decision-making to every individual.

### 2.3. Paradigm Shift from Curation to Prevention

The third development that requires a different focus and organisation of paediatric care concerns the paradigm shift from curation to prevention. This paradigm shift is related to the increasing number of children with chronic conditions because of an unhealthy lifestyle and the reduced incidence of acute illnesses that can be cured. The fact that chronic diseases cannot be cured, but can be prevented, requires a different approach from paediatricians: instead of treating the consequences of a poor lifestyle, the paediatrician is more and more forced and motivated to stress the importance of the benefits of a healthy lifestyle, and to discuss and address the drawbacks and risks of an unhealthy lifestyle. This paradigm shift is especially essential for children: a large investment in prevention at a young age contributes to improved health of children now and in their future.

## 3. Implementation of Building blocks

To be able to anticipate the developments mentioned above, which are expected to continue for some years of even decades, the Dutch Paediatric Society has developed a renewed vision. It is a vision towards the year 2030 that gives children and paediatric care the right place in the Dutch healthcare landscape to guarantee accessible, high-quality, and effective care for every child at the right time. In the following text, we indicate which building blocks should be implemented to realise this vision.

### 3.1. Building Block 1: A Solid Foundation of Paediatric Care

In 2030, Dutch paediatricians strive as much as possible to make joint decisions with the young patient and/or its family and to make them as comfortable as possible. Specific attention will be paid to children and families with limited health literacy and/or social network. The pediatrician organises the provision of information and professional support with these vulnerable families in such a way that it encourages and facilitates shared decision-making. He/she will assess the level of knowledge and the competences of the young patient and decision-making processes will be adapted to the child’s competences. In addition, in determining what is important for health, more than before there will be attention to feelings and thoughts, now and later, feeling comfortable and participating in daily life, important elements of the concept of positive health [18]. A broader view of paediatricians on the concept of health means that they will have a more signaling role in case of a discrepancy between factors and demands from a family’s environment that cause stress and their skills and capacity to cope with these stressful situations. It also requires that the pediatrician has access to a wide network with expertise in different fields. Making use of this ensures that not the patient and its family has to go wherever the expertise is, but that the expertise visits the child.

### 3.2. Building Block 2: Interprofessional Collaboration between Paediatricians and Other Health Care Providers

In 2030, Dutch paediatricians will work together in networks within and outside the medical domain. A broader view on health, including physical, mental and social well-being, will ask for more connection between paediatrics and other domains of care, such as prevention and social care. The paediatrician can stimulate and initiate an interdisciplinary network around the child in which the pediatrician, depending on the dominance of the medical problem, plays a more of a less prominent role. There is no question of a fixed network, but of a flexible network in which the disciplines involved align the needs of the specific child. In this network, it is clear what role the pediatrician takes (e.g., the main health care provider or coordinator of care), which other roles are desired and to whom these should be assigned. Pediatricians actively invest in connecting, establishing, and further developing professional networks (including patient societies) that enable collaboration within and outside the medical domain.

### 3.3. Building Block 3: Local and Accessible Pediatric Care

In 2030, paediatric knowledge, support and care will always be nearby the child and its family in The Netherlands. To achieve this, hybrid care, whereby the child and the family decide on a physical or online consultation, will become the norm. In this way the paediatrician can work ‘place independent’, which guarantees proximity and availability of care. By making use of the aforementioned multidisciplinary networks, also virtual, knowledge exchange, contact and interprofessional telephone/internet consultation can take place quickly and easily. This will also account for regional acute paediatric care: paediatric expertise will come to the patient instead of the other way around that the child and the family has to travel to the expertise. To realise accessible acute paediatric care of high-quality innovative ways of transportation will be implemented such as the “white roadside assistance”: a highly qualified and experienced team of professionals will arrive on sight to provide paediatric care in acute medical situations.

### 3.4. Building Block 4: Concentration of High-Complex, Low-Frequency Pediatric Care

In 2030, every child and family with their highly complex, low-frequency care needs can go to the place where the best care is offered for them. To be able to offer the right expertise for highly complex, low-frequency care questions, concentration in one or a few (national) centers of expertise is necessary. A good example of concentration of high-complex, low-frequency pediatric care is the Princess Máxima Center for Pediatric Oncology in Utrecht. The center provides pediatric oncological care to all Dutch children with cancer. E-health, home monitoring and other technological innovations can also be used within these specialised centers, among other things to limit the travel of the child and family.

As indicated earlier, implementation of the building blocks is intended to anticipate demographic, epidemiological and social developments that will influence the organisation of pediatric care in The Netherlands. By investing in the coming decade in (1) strengthening the role of the child and the family in the care process, (2) making care more accessible, for example through e-health, but also by bringing expertise to the patient instead of the other way around, (3) broadening and strengthening the network around the pediatrician and in (4) the concentration of concentration of high-complex, low-frequency pediatric care, the Dutch Paediatric Society hopes to continue to guarantee access to and quality of paediatric care.

## 4. Who and What Is Needed to Realise This Renewed Vision?

The implementation of building blocks requires adjustments elsewhere in the healthcare system. We want to argue for several preconditions that should be met to implement the building blocks and with that to improve access to and quality of paediatric care. A first requisite is a joint electronic patient file. A patient file in which all diagnostics, treatment plans and positive health aspects of the child are collected, and which is accessible to all professionals involved. It guarantees a barrier-free exchange of medical and non-medical information between healthcare professionals across the domains and within the framework of the current privacy legislation. 

The second precondition is reviewing healthcare financing. The financing should facilitate transcending actions in care and support for every child and its family. This can be achieved by decommissioning and simplification of funding in which the child is central.

The last precondition concerns an adjustment in the paediatricic training curriculum. The paediatrician of the future should have knowledge of positive health and integrative medicine. He or she has to be trained to function in networks and to make connections. The future paediatrician must also be trained to take into account differences in health skills of the child and family.

## 5. Conclusions

By renewing their vision, the Dutch Paediatric Society anticipates developments that the health care system can no longer ignore. This requires not only an adjustment from paediatricians and paediatricians in training, but also from professionals who work with them and from the Government that can facilitate or accelerate the implementation of the renewed vision in various ways. 

## Figures and Tables

**Table 1 ijerph-19-01037-t001:** Indicators of child health and Dutch numbers (2020).

Indicator of Child Health	Dutch Numbers
Mortality rate per 1000 live born children [3]	
Neonatal	2.88
Infant	3.81
Under-five	4.28
Age 5–14	0.75
Age 15–19	0.91
Preterm births [4]	6.9%
Caesarean sections [5]	17.3%
N of paediatric hospitals [6]	7
N of paediatricians in the country	1800
MMR vaccination rate [7]	93.6%
N of child abuse reports [8]	62.470
N of children receiving youth care [9]	429.200

## Data Availability

Not applicable.

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
