# Peer review of "Paediatric Care in The Netherlands: State of Affairs, Challenges and Prospects"

_ijerph, 2022, doi:10.3390/ijerph19031037_

Round 1

Reviewer 1 Report

I read with interest this article, which introduces a special issue on the Health of children in the Netherlands.

I think it tackles an important issue for a post-pandemic vision of future development of a European advanced health-care system, as the Dutch one.

Introduction

I think the introduction could benefit from the inclusion of additional information regarding the source of data reported, as the website reported in the references [2] is available only in Dutch language.
How are the data collected? Which is the source/s of information used?

I consider very informative the inclusion of a table in which some major indicators (see below) of child health could be reported in order to have a snapshot of the Dutch situation and to allow comparability with other countries.

Natimortality (N of fetal deaths per 1000 newborns)

Neonatal mortality ratio

Infant mortality rate

Pediatric mortality rate

Preterm birth rate

Caesarian section rate

Estimated N of children with special health care needs

Estimated N of children with RD

Hospitalization rate

It would be important to report also the N of pediatric hospitals

the N of pediatricians in the country

and the N of children resident per paediatrician.

Are there differences in these numbers per country regions?

I think other key-areas important for children health are vaccination and the issue of child abuse. They should be described in the article.  

The Authors tackled the issue of the growing number of children with chronic and sometimes severe conditions. I am not fully aware of the state of the art of the debate on assisted suicide in children, but I think that the position of the Dutch Society should be reported.

The readers could be interested in knowing if some data are available on this and which is the role of the pediatrician in this process. Moreover, which are the future directions, if any?

No major comment on the part regarding building blocks, apart from a general consideration. How to take into account the changes in child competencies in the care process? i.e. in terms of knowledge of their disease, of technological competencies, etc.?

Minor comments

Line 88: please add “parents and guardians health literacy”

Line 197: around the patient rather than “around the pediatrician”

Author Response

Response to Reviewer 1 Comments

Point 1: I read with interest this article, which introduces a special issue on the Health of children in the Netherlands. I think it tackles an important issue for a post-pandemic vision of future development of a European advanced health-care system, as the Dutch one.

Response 1: thank you very much for your positive response.

Point 2: I think the introduction could benefit from the inclusion of additional information regarding the source of data reported, as the website reported in the references [2] is available only in Dutch language. How are the data collected? Which is the source/s of information used?

Response 2: Thank you for this question. We have provided more information on the data source. We have added the text at line 31-33 (added text in Italics):

According of the website of the Dutch Healthcare Authority, which provides care and treatment data supplied by Dutch hospitals (e.g., how many patients are diagnosed, the associated care products and care activities), this number is lower than previous years (452.138 in 2019 and 447.081 in 2018) because of the COVID-19 pandemic [2].”

In addition, for the convenience of the reader, we have translated the Dutch sources in the reference list into English so that they are more informative.

Point 3: I consider very informative the inclusion of a table in which some major indicators (see below) of child health could be reported in order to have a snapshot of the Dutch situation and to allow comparability with other countries.

Natimortality (N of fetal deaths per 1000 newborns)

Neonatal mortality ratio

Infant mortality rate

Pediatric mortality rate

Preterm birth rate

Caesarian section rate

Estimated N of children with special health care needs

Estimated N of children with RD

Hospitalization rate

It would be important to report also the N of pediatric hospitals

the N of pediatricians in the country

and the N of children resident per paediatrician.

Are there differences in these numbers per country regions?

Response 3: thank you for this very interesting suggestion. We have added the requested information that is available for the Netherlands in a Table:

Table 1: Indicators of child health and Dutch numbers (2020)

Indicator of child health

Dutch numbers

Mortality rate per 1,000 live born children1

Neonatal

2.88

Infant

3.81

Under-five

4.28

Age 5-14

0.75

Age 15-19

0.91

Preterm births2

6,9%

Caesarean sections3

17,3%

N of paediatric hospitals4

7

N of paediatricians in the country

1800

MMR vaccination rate5

93,6%

N of child abuse reports6

62.470

N of children receiving youth care7

429.200

1 https://childmortality.org/data/Netherlands

2 https://www.volksgezondheidenzorg.info/onderwerp/vroeggeboorte-ondergewicht-enof-groeivertraging/cijfers-context/huidige-situatie - !node-vroeggeboorten-naar-zwangerschapsduur

3 https://www.volksgezondheidenzorg.info/onderwerp/zorg-rond-de-geboorte/cijfers-context/gebruik#node-bevallingen-naar-wijze-van-bevallen

4 https://www.volksgezondheidenzorg.info/onderwerp/ziekenhuiszorg/cijfers-context/aanbod#node-aantal-instellingen-voor-medisch-specialistische-zorg

5 https://rijksvaccinatieprogramma.nl/nieuws/rapport-ontwikkelingen-rijksvaccinatieprogramma-2019-landelijke-vaccinatiegraad-gestegen

6 https://www.nji.nl/cijfers/veilig-thuis

7 https://www.cbs.nl/nl-nl/nieuws/2021/17/daling-aantal-jongeren-met-jeugdzorg-in-2020

Point 4: I think other key-areas important for children health are vaccination and the issue of child abuse. They should be described in the article.

Response 4: Thank you for this suggestion. We agree that vaccination and child abuse are important indicators of child health. We have added figures in Table 1.

Point 5: The Authors tackled the issue of the growing number of children with chronic and sometimes severe conditions. I am not fully aware of the state of the art of the debate on assisted suicide in children, but I think that the position of the Dutch Society should be reported. The readers could be interested in knowing if some data are available on this and which is the role of the pediatrician in this process. Moreover, which are the future directions, if any?

Response 5: The Dutch Paediatric Society does not have an explicit policy on assisted suicide. In the Netherlands we have a well-organized network for child and adolescent psychiatrists that deal with these kinds of issues. In this article we can therefore not describe the role of the paediatrician in this, nor are we able to describe future directions.

Point 6: No major comment on the part regarding building blocks, apart from a general consideration. How to take into account the changes in child competencies in the care process? i.e. in terms of knowledge of their disease, of technological competencies, etc.?

Response 6: Thank you for this valuable consideration. In the societal developments we indeed described the changing role of the paediatric patient in the care process. In line 147-149 we added more text to describe the role of the paediatrician re this consideration:

“He/she will assess the level of knowledge and the competences of the young patient and decision-making processes will be adapted to the child's competences.”

Point 7: Minor comments

Line 88: please add “parents and guardians health literacy”

Response 7: We added this.

Point 8: Line 197: around the patient rather than “around the pediatrician”

Response 8: We did not change this because we actually mean “around the paediatrician”. To improve provided care, the paediatrician should collaborate more with other health professionals and will therefore have to expand the network.

Reviewer 2 Report

The unquestionable strength of the paper is that while it describes the challenges and prospects the paediatric care in the Netherlands will face in the future in fact it could be applied to many other Western societies. For example, the Authors aptly indicate three main areas that affect a rising and changing demand for paediatric health care in the Netherlands: an aging population, more diversity in cultural background and a rising prevalence of chronic diseases and multi morbidity among children and adolescents. Moreover, because the Duch health care system is one of the best in Europe some ideas sketch by the Authors could serve as a good starting point in the discussion on the future of paediatric care in various  countries in the UE. For example, one must agree that especially in the field of paediatric care there is a urgent need for paradigm shift from curation to prevention. The Authors also suggest some ‘building blocks’ that should be implemented in order to optimize the developments paediatric care in the county. Finally, apart from stressing the need for reviewing healthcare financing the Authors aptly suggest that improving paediatric care also requires investing in  electronic patient record and paediatric training. To conclude, I believe that the issues raised in this viewpoint are important and timely and for that reason it can be accepted for publication in the Journal. I have only one minor comment/suggestion: while describing the rising prevalence of chronic diseases and multimorbidity among children and adolescents I  would expect some more information on the demand for psychiatric care among children and adolescents, which result from increasing problems with depression, various type of addictions or suicidal ideations and/or behaviours.

Author Response

Response to Reviewer 2 Comments

Point 1: The unquestionable strength of the paper is that while it describes the challenges and prospects the paediatric care in the Netherlands will face in the future in fact it could be applied to many other Western societies. For example, the Authors aptly indicate three main areas that affect a rising and changing demand for paediatric health care in the Netherlands: an aging population, more diversity in cultural background and a rising prevalence of chronic diseases and multi morbidity among children and adolescents. Moreover, because the Duch health care system is one of the best in Europe some ideas sketch by the Authors could serve as a good starting point in the discussion on the future of paediatric care in various  countries in the UE. For example, one must agree that especially in the field of paediatric care there is a urgent need for paradigm shift from curation to prevention. The Authors also suggest some ‘building blocks’ that should be implemented in order to optimize the developments paediatric care in the county. Finally, apart from stressing the need for reviewing healthcare financing the Authors aptly suggest that improving paediatric care also requires investing in  electronic patient record and paediatric training. To conclude, I believe that the issues raised in this viewpoint are important and timely and for that reason it can be accepted for publication in the Journal. I have only one minor comment/suggestion: while describing the rising prevalence of chronic diseases and multimorbidity among children and adolescents I  would expect some more information on the demand for psychiatric care among children and adolescents, which result from increasing problems with depression, various type of addictions or suicidal ideations and/or behaviours.

Response 1: Thank you very much for these praising words. We have added a table with some background information at the request of the other reviewer, see below. We have added some figures about the demand for youth care.

Table 1: Indicators of child health and Dutch numbers (2020)

Indicator of child health

Dutch numbers

Mortality rate per 1,000 live born children1

Neonatal

2.88

Infant

3.81

Under-five

4.28

Age 5-14

0.75

Age 15-19

0.91

Preterm births2

6,9%

Caesarean sections3

17,3%

N of paediatric hospitals4

7

N of paediatricians in the country

1800

MMR vaccination rate5

93,6%

N of child abuse reports6

62.470

N of children receiving youth care7

429.200

1 https://childmortality.org/data/Netherlands

2 https://www.volksgezondheidenzorg.info/onderwerp/vroeggeboorte-ondergewicht-enof-groeivertraging/cijfers-context/huidige-situatie - !node-vroeggeboorten-naar-zwangerschapsduur

3 https://www.volksgezondheidenzorg.info/onderwerp/zorg-rond-de-geboorte/cijfers-context/gebruik#node-bevallingen-naar-wijze-van-bevallen

4 https://www.volksgezondheidenzorg.info/onderwerp/ziekenhuiszorg/cijfers-context/aanbod#node-aantal-instellingen-voor-medisch-specialistische-zorg

5 https://rijksvaccinatieprogramma.nl/nieuws/rapport-ontwikkelingen-rijksvaccinatieprogramma-2019-landelijke-vaccinatiegraad-gestegen

6 https://www.nji.nl/cijfers/veilig-thuis

7 https://www.cbs.nl/nl-nl/nieuws/2021/17/daling-aantal-jongeren-met-jeugdzorg-in-2020